# A Novel Hydroxyapatite/Vitamin B_12_ Nanoformula for Treatment of Bone Damage: Preparation, Characterization, and Anti-Arthritic, Anti-Inflammatory, and Antioxidant Activities in Chemically Induced Arthritic Rats

**DOI:** 10.3390/ph16040551

**Published:** 2023-04-06

**Authors:** Amany Belal, Rehab Mahmoud, Eman E. Mohamed, Ahmed Farghali, Fatma I. Abo El-Ela, Amr Gamal, Fatma Mohamed Halfaya, Esraa Khaled, Abdelbasset A. Farahat, Ahmed H. E. Hassan, Mohammed M. Ghoneim, Mohamed Taha, Mohamed Y. Zaky

**Affiliations:** 1Department of Pharmaceutical Chemistry, College of Pharmacy, Taif University, Taif 21944, Saudi Arabia; 2Department of Chemistry, Faculty of Science, Beni-Suef University, Beni-Suef 62511, Egypt; 3Molecular Physiology Division, Zoology Department, Faculty of Science, Beni-Suef University, Beni-Suef 62511, Egypt; 4Materials Science and Nanotechnology Department, Faculty of Postgraduate Studies for Advanced Sciences, Beni-Suef University, Beni-Suef 62511, Egypt; 5Department of Pharmacology, Faculty of Veterinary Medicine, Beni-Suef University, Beni-Suef 62511, Egypt; 6Department of Pharmaceutics and Industrial Pharmacy, Faculty of Pharmacy, Beni-Suef University, Beni-Suef 62511, Egypt; 7Department of Surgery, Anesthesiology and Radiology, Faculty of Veterinary Medicine, Beni-Suef University, Beni-Suef 62511, Egypt; 8Master of Pharmaceutical Sciences Program, California Northstate University, 9700 W Taron Dr., Elk Grove, CA 95757, USA; 9Department of Pharmaceutical Organic Chemistry, Faculty of Pharmacy, Mansoura University, Mansoura 35516, Egypt; 10Department of Medicinal Chemistry, Faculty of Pharmacy, Mansoura University, Mansoura 35516, Egypt; 11Medicinal Chemistry Laboratory, College of Pharmacy, Kyung Hee University, 26 Kyungheedae-ro, Seoul 02447, Republic of Korea; 12Department of Pharmacy Practice, College of Pharmacy, AlMaarefa University, Ad Diriyah 13713, Saudi Arabia; 13Department of Oncology and Department of Biomedical and Clinical Sciences, Faculty of Medicine, Linköping University, 581 83 Linköping, Sweden

**Keywords:** bone damage, bone disability research, rheumatoid arthritis, nano-HAP/Vit B_12_, anti-arthritic, antioxidant, anti-inflammatory

## Abstract

The usage of nanomaterials for rheumatoid arthritis (RA) treatment can improve bioavailability and enable selective targeting. The current study prepares and evaluates the in vivo biological effects of a novel hydroxyapatite/vitamin B_12_ nanoformula in Complete Freund’s adjuvant-induced arthritis in rats. The synthesized nanoformula was characterized using XRD, FTIR, BET analysis, HERTEM, SEM, particle size, and zeta potential. We synthesized pure HAP NPs with 71.01% loading weight percentages of Vit B12 and 49 mg/g loading capacity. Loading of vitamin B_12_ on hydroxyapatite was modeled by Monte Carlo simulation. Anti-arthritic, anti-inflammatory, and antioxidant effects of the prepared nanoformula were assessed. Treated arthritic rats showed lower levels of RF and CRP, IL-1β, TNF-α, IL-17, and ADAMTS-5, but higher IL-4 and TIMP-3 levels. In addition, the prepared nanoformula increased GSH content and GST antioxidant activity while decreasing LPO levels. Furthermore, it reduced the expression of TGF-β mRNA. Histopathological examinations revealed an improvement in joint injuries through the reduction of inflammatory cell infiltration, cartilage deterioration, and bone damage caused by Complete Freund’s adjuvant. These findings indicate that the anti-arthritic, antioxidant, and anti-inflammatory properties of the prepared nanoformula could be useful for the development of new anti-arthritic treatments.

## 1. Introduction

Rheumatoid arthritis (RA) is an inflammatory disease characterized by chronic inflammation and destruction of bone and cartilage in the ankle joint, accompanied by swelling and pain, ultimately leading to joint deformation [1,2]. RA triggers recurrent synovitis, the production of autoantibodies, and bone and cartilage destruction of the ankle joint, as well as chronic pain and impaired joint function [3]. RA affects approximately 1–2% of the global population, with onset usually in people’s 40s and 50s, though onset is possible at any age [4]. The specific etiology of RA is still unknown. However, it has been demonstrated that a range of effector cell activities may contribute to RA pathogenesis [5,6].

Reactive oxygen species (ROS) contribute to the pathophysiology of RA and several other diseases [7,8]. Evidence of joint inflammation and degradation because of oxidative stress was found in arthritis animal models and in RA patients [9]. In arthritic animals, levels of ROS indicators of protein and lipid oxidation are elevated. Moreover, oxidative status changes have been observed in RA patients’ serum, as well as in experimentally arthritis-induced rats [10]. Several serum growth factors, including chemokines and cytokines, have been implicated in RA development. Because of their roles in promoting inflammation and other biological processes, cytokines are crucial for RA progression. In rheumatic joints, pro-inflammatory/anti-inflammatory cytokine imbalance leads to an autoimmune response that destroys joints [11].

Currently, some disease-modifying antirheumatic drugs in combination with physiotherapy retard the progression of RA [12]. Nonsteroidal and steroidal anti-inflammatory medications are used as first-line therapy for RA [13]. Unfortunately, an effective novel RA treatment is still required.

The application of nanotechnology in medical treatment is a hot research area. Such a therapeutic strategy would be applied in alternative and integrative medicine and would accelerate the development of treatments for several pathological disorders, including RA [14]. Hydroxyapatite (HAP), an inorganic material, has good biocompatibility, bioactivity, and safety. HAP can serve as a nanodrug delivery carrier, as it degrades effectively under acidic conditions [15]. HAP has been used for the sustained delivery of antibiotics and drugs for tissue regeneration and cell differentiation [16]. Nanohydroxyapatite (nano-HAP) has a beneficial outcome on biomaterial biocompatibility and bioactivity [17]. Minimal tissue reaction and more pronounced activity are achieved upon using nano-HAP, as it has a larger surface area relative to micro-HAP [18]. Nano-HAP can permeate through osseous tissue and deliver calcium phosphates required for bone regeneration after disease injury [3,19,20]. Consequently, Vit B_12_-loaded HAP might afford a nanoformula useful for RA treatment. In this lieu, the current work addressed the assessment of the anti-arthritic effects of nano-HAP/Vit B_12_ in rats with CFA-induced arthritis considering anti-inflammatory and antioxidant activities, as well as ADAMTS5, TIMP3, and TGF-β roles.

## 2. Results

### 2.1. Characterization of Nano-HAP and Nano-HAP/Vit B_12_

The XRD of the prepared samples is illustrated in Figure 1A. Nano-HAP showed typical XRD peaks for the HAP phase. Peaks at 25.9°, 31.6°, 33°, 34°, 39.3°, 46.7°, 47.9°, 49.7°, 53.2°, and 64° can be indexed to the plane families (002), (211), (300), (202), (212), (222), (132), (213), (004), and (323) that support the discovery of an admixture phase identified as Ca4Mg5(PO4)6 calcium magnesium phosphate (ICDD Card Number: 00-011-0231). According to ICDD Card Number 01-073-0293, HAP crystallized in a hexagonal form. No significant difference was observed in XRD pattern peaks of loaded HAP-Vit B_12_, as increasing some peak intensities such as plane (211), peak shift, and despacing decreases after Vit B_12_ loading, as shown in Appendix A.

FTIR analysis of the prepared samples is shown in Figure 1B. Absorption bands at 3450 cm^−1^ and 1648 cm^−1^ correspond to hydroxyl groups’ stretching and bending. Because of carbon dioxide absorption during the synthesis of nano-HAP in alkaline media, bands corresponding to carbonate ions are visible at 1430 cm^−1^ and 869 cm^−1^. Bands at 1045 cm^−1^ and 575 cm^−1^ are typical peaks for the asymmetric stretching and bending of PO_4_^−3^ ions, respectively. The presence of dipole–dipole hydrogen bonding is confirmed by FTIR, which showed a decline in the intensity and shifting of the OH group band toward the minimally higher wavenumbers at approximately 3423.85 cm^−1^. Additionally, the intensities of OH and PO_4_^−3^ bands at 1640.14 and 1035.78 cm^−1^ are substantially reduced and shifted to 1643.03 and 1031.14 cm^−1^ in nano-HAP/Vit B_12_, indicating the existence of a coordination mechanism between the two substances (Figure 1B). The signed spectra of Vit B_12_ (Figure 1B) are at 3420.05, 2974.93, 2374.66, 1801.08, 1420.70, 873.93, and 708.90 cm^−1^. The band assigned at 2974.93 cm^−1^ is characteristic of stretching of C-H, and the band assigned at 1801.01 cm^−1^ is characteristic of stretching of C=C, 1420.70 cm^−1^ for C=O, and 873.93 and 709.90 cm^−1^ for C-O. The appearance of some peaks related to Vit B_12_ in HAP/Vit B_12_ such as 1432.87 and 875.13 cm^−1^ conforms to the loading of Vit B_12_. Additionally, broadening in the peak at 3421.63 cm^−1^ conforms to the increase in the H-bond.

The N_2_ adsorption–desorption isotherms of nano-HAP and nano-HAP/Vit B_12_ samples are shown in Figure 1C. With a type H3 hysteresis loop, both isotherms fall under the type IV category. They have a pore size up to 55 nm, which are likely intra-particle pores formed upon nanoparticle aggregation, as indicated by the inset’s pore size distribution. BET surface areas for nano-HAP and nano-HAP/Vit B_12_ were 83.6 and 73.9 m^2^/g, respectively. The surface area decreases after the loading of Vit B_12_ confirms its adsorption. This was consistent with SEM observations (Figure 1D) and supported by the wide distribution of pore size (Appendix A). Specific surface areas (BET analysis) were 67.184 and 55.816 m^2^/g, while maximum pore volumes were about 0.48918 and 0.476947 cc/g for nano-HAP and nano-HAP/Vit B_12_, respectively. The specific surface area results from the porous structure of nanoparticles (Figure 1C,D). Such a porous structure is favorable for medical applications. Hydrodynamic (HD) sizes of nano-HAP and nano-HAP/Vit B_12_ are shown in Figure 1E. Nano-HAP has a large HD of about 665 nm, which indicates particle agglomeration in solution. On the other hand, nano-HAP/Vit B_12_ had a smaller HD of about 312 nm, which can be attributed to the capping action by Vit B_12_ that inhibited particle agglomeration. Both nano-HAP and nano-HAP/Vit B_12_ showed an insignificant zeta potential (almost zero mV), indicating uncharged surfaces prior to and post-Vit B_12_ loading and, hence, the null capability for electrostatic attraction of counters ions in solution.

As shown in Figure 2A, scanning electron microscopy (SEM) revealed variable sizes and positions and a consistently rough surface with numerous pores. Collections of rod-shaped nanoparticles were obvious and piled on each other. The observed morphological porosity was in good agreement with the BET study. Two dominant morphologies were clear. The first is layered-type aggregates forming 2D structures. The other is also a layered structure but with much shorter widths than their length and, hence, forming 2D rod-like aggregates. EDX analysis (Figure 2B) reflected the purity of the sample with dominant peaks originating from calcium and phosphorous in addition to a weak sodium peak, probably originating from sodium hydroxide traces. In the nano-HAP/Vit B_12_ TEM image (Figure 2C), both layered and rod morphologies were visible. Layered structures showed width sizes ranging from 45 to 125 nm, while rods had much shorter widths of about 25 nm. Meanwhile, some large layers were formed (Figure 2D) with a width of about 800 nm and a length of 1660 nm (1.6 µm).

### 2.2. Molecular Modeling of HAP/Vit B_12_

Monte Carlo (MC) simulation was addressed to understand and visualize the loading mechanism of Vit B_12_ on the (0 1 0) HAP surface. The calculated adsorption energy was 121 kcal/mol, indicating favorable interactions between Vit B_12_ and the HAP surface. The lowest-energy structure of the adsorbed Vit B_12_ is displayed in Figure 3. The interaction between Vit B_12_ and the (0 1 0) HAP surface was through hydrogen bond formation between the hydrogens of the Vit B_12_ amide groups and oxygens of the HAP surface, as well as electrostatic interactions between Vit B_12_ oxygen atoms and HAP calcium atoms.

### 2.3. Effect of Nano-HAP/Vit B_12_ on Gross Lesions of the Paw and Ankle Joints

An in vivo arthritis model was established to evaluate the effects of nano-HAP/Vit B_12_ on CFA-induced arthritis. Compared with the negative (normal) control, the right hind paws and ankles of the positive control (CFA-induced arthritic rats) showed clear swelling, edema, and redness (Figure 4A,B), which are external signs of arthritis. Arthritic rats that received nano-HAP/Vit B_12_ had significantly fewer visible lesions (Figure 4C). These findings suggest anti-inflammatory effects of nano-HAP/Vitamin B_12_.

### 2.4. Effect of Nano-HAP/Vit B_12_ on Right Hind Paws Volume

Digital calipers were used to assess paw edema in various groups, as well as the diameter of the right hind leg in the paw region to assess swelling rates. Compared with the negative control group, there was a substantial (*p* ˂ 0.05) increase in paw size (edema) of CFA-induced arthritic rats (Figure 5). Meanwhile, nano-HAP/Vit B_12_-treated rats had less swollen paws relative to the positive control group.

### 2.5. Effect of Nano-HAP/Vit B_12_ on LPO, GSH Content, and GST Activity

Nano-HAP/Vit B_12_ administration significantly (*p* < 0.05) reduced LPO levels and increased GSH content and GST activity in comparison to the positive control group, which had significantly (*p* < 0.05) higher LPO levels and lower GSH content and GST activity than negative control rats (Figure 6A–C).

### 2.6. Effects of Nano-HAP/Vit B_12_ on Serum RF, CRP, TNF-α, IL-1β, IL-17, IL-4, ADAMTS-5 and TIMP3 Levels

The effect of nano-HAP/Vit B_12_ on RF, CRP, TNF-α, IL-1β, IL-17, IL-4, ADAMTS-5, and TIMP3 levels is depicted in Figure 7A–H. CFA administration increased (*p* ˂ 0.05) the serum levels of RF, CRP, TNF-α, IL-1β, IL-17, and ADAMTS-5, as well as decreasing (*p* ˂ 0.05) the serum levels of IL-4 and TIMP3. Nano-HAP/Vit B_12_-treated arthritic rats showed significantly (*p* ˂ 0.05) lower serum levels of RF, CRP, TNF-α, IL-1β, and IL-17 relative to the positive group. Meanwhile, nano-HAP/Vit B_12_ treatment significantly (*p* ˂ 0.05) increased IL-4 and TIMP-3 serum levels compared with the positive group.

### 2.7. Effect of Nano-HAP/Vit B_12_ on TGF-β mRNA Expression

Rats receiving CFA had considerably (*p* < 0.05) higher TGF-β mRNA expression than normal controls rats. Such increased TGF-β expression in arthritic rats was significantly (*p* < 0.05) downregulated in nano-HAP/Vit B_12_-treated rats (Figure 8).

### 2.8. Histopathological Changes and Arthritic Score

Histopathological examination demonstrated no inflammation in the right ankle joint sections of normal control rats’ hind leg (Figure 9A). Conversely, the synovium of CFA-induced arthritic rats showed hyperplasia, substantial inflammatory cell infiltration, and substantial cartilage destruction (Figure 9B). Differently, nano-HAP/Vit B_12_-treated rats possessed nearly normal articular surfaces and synovial membranes free from inflammation (Figure 9C). In addition, nano-HAP/Vit B_12_ reduced cartilage deterioration, suppressed pannus development, and decreased synovitis. The joint damage’s histology lesion scores are displayed in Table 1.

## 3. Discussion

XRD (Figure 1A) showed that nano-HAP/Vit B_12_ retained the typical diffraction peaks of the nano-HAP structure [21]. No significant difference was observed in XRD pattern peaks with the increasing intensities of some peaks and shifted peak positions and decreased despacing after the adsorption of Vit B12 (Appendix A).

FTIR analysis (Figure 1B) featured absorption bands for the stretching and bending vibration of hydroxyl groups at 3450 cm^−1^ and 1648 cm^−1^ [22,23]. Bands at 1430 cm^−1^ and 869 cm^−1^ indicated the existence of carbonate ions due to carbon dioxide adsorption during the synthesis of nano-HAP from alkaline media [24]. Bands at 1045 cm^−1^ and 575 cm^−1^ are typical for the asymmetric stretching and bending of PO_4_^−3^ ions, respectively [25].

There was no discernible difference between nano-HAP and nano-HAP/Vit B_12_ in terms of pore size distribution. This indicates that Vit B_12_ was adsorbed on the surface of HAP nanoparticles rather than inside the pores [26]. The decrease in surface area post-Vit B_12_ loading confirmed adsorption of Vit B_12_. This result was consistent with the SEM observations (Figure 1D). In the SEM pictures, various sizes and positions were observed, which showed a consistently rough surface with numerous pores [27].

To explore the therapeutic potential of nano-HAP/Vit B_12_ on CFA-induced arthritis, an in vivo arthritis model was used. To evaluate the anti-inflammatory effects of RA, paw edema was measured [28]. The findings revealed that the treatment of arthritic rats with nano-HAP/Vit B_12_ reduced the size of the swollen hind paw compared with arthritic rats. Such a paw size reduction indicates a slower swelling rate, which may be due to the suppression of inflammatory processes [29]. This might be attributed to nano-HAP/Vitamin B_12_’s anti-inflammatory effects.

Oxidative stress occurs when oxidant production exceeds the cell’s antioxidant defenses, resulting in oxidative damage to the cell. Oxidative stress plays a role in the etiology of rheumatoid arthritis (RA) and other diseases. Increased reactive oxygen species (ROS) concentrations beyond the physiological standards induce oxidative stress [30]. ROS are unique signal mediators that play vital roles in cell growth and differentiation [31]. Oxidative stress triggers free radical cytotoxicity, which is associated with several diseases [32]. The results demonstrated that CFA-induced arthritic rats had significantly higher LPO levels and lower GSH content and GST activity. These data are consistent with Ahmed et al. [33] and Saleem et al. [34]. An elevated level of LPO indicates the production of ROS. In arthritic rats, excessive ROS production may have lowered the antioxidant defense system, increased LPO levels, and rendered synovial fluid and collagen susceptible to oxygen radicals-mediated damage [35].

The administration of nano-HAP/Vit B_12_ to arthritic rats resulted in a significant increase in GSH content and GST activity, as well as a reduction of LPO levels to near negative control group levels, indicating scavenging of ROS or antioxidant potential, implying the effectiveness of nano-HAP/Vit B_12_ for the treatment of RA. These results are in parallel with Ain et al. [3]. Accordingly, it may be deduced that oxidative stress is one of the primary mechanisms for inhibiting inflammatory cytokine in RA [34]. Additionally, it has been shown that individuals with RA encounter significant levels of oxidative stress in addition to elevated inflammation [36]. These findings suggested that the nano-HAP/Vit B_12_ had a strong antioxidant activity against ROS because of CFA arthritis induction.

CRP and RF are markers of systemic inflammation and the generation of antibodies against RA [37]. In parallel with Yang et al. [38], serum RF and CRP concentrations were significantly higher in arthritic rats than in normal rats. Serum RF and CRP were used to evaluate joint activity in RA-affected rats as they are two key indicators of systemic inflammation in RA [39]. Treating arthritic rats with nano-HAP/Vit B_12_ significantly reduced both RF and CRP serum levels, suggesting that nano-HAP /Vit B_12_ has anti-inflammatory effects. Several studies have shown that inflammation is a primary mechanism and an important function in RA rats [40,41,42,43]. Consistent with our findings, the arthritic positive control group revealed significantly increased serum levels of IL-1β, TNF-α, and IL-17 and a significantly decreased level of IL-4 in arthritic rats. Our results are in harmony with Yang et al. [38], Li et al. [44], and Setiadi and Karmawan [45]. Additionally, synovial inflammation and cartilage destruction are caused by the overproduction of pro-inflammatory cytokines such as TNF-α, IL-1β, and IL-17, as well as decreased anti-inflammatory factors such as IL-4, which have been associated with RA [40,41,42,43]. In RA, TNF-α, IL-1β, and IL-17 all play significant and cooperative roles in cartilage degradation and synovial inflammation [34,40,46]. TNF-α overproduction also results in the production of matrix-degrading enzymes and higher amounts of IL-1β [40]. IL-1β is the most crucial cytokine in the formation of pathogenic arthritis and has been connected to signs such as morning stiffness. This cytokine, which is mainly produced by macrophages, has a vital role in the invasion of inflammatory cells, as well as the deterioration of cartilage and bone [47]. Additionally, IL-17 is essential for RA development, as it induces the overproduction of pro-inflammatory cytokines, osteoclast activation, and angiogenesis [40]. According to these data, therapeutic agents that effectively reduce the release of TNF-α, IL-1β, IL-6, and IL-17 could be beneficial RA treatments [40,46,48]. IL-4 is essential for regulating the levels of endogenous pro-inflammatory cytokines during RA [34,40].

According to studies, RA patients have higher levels of several cytokines than healthy individuals. These cytokines are released by immune cells to enhance cell interaction during inflammation. Since these cytokines are RA modulators, current standard RA treatments aim to reduce inflammation by inhibiting their release [49,50]. According to our data, nano-HAP/Vit B_12_ treatment obviously reduced TNF-α, IL-1β, and IL-17 levels and elevated IL-4 expression, suggesting that the anti-inflammatory impact of nano-HAP/Vit B_12_ might happen through the suppression of pro-inflammatory cytokines and the elevation of anti-inflammatory cytokines. Therefore, anti-rheumatic drug delivery using nano-HAP has received attention. HAP is an efficient carrier for RA due to its anti-inflammatory [51,52] and osteoinductive characteristics [53,54].

ADAMTS-5 is the main aggrecanase recently discovered and has been found to be expressed in a variety of tissues, including cartilage. It is reported that the ADAMTS-5 level is regulated by many metabolic factors, such as TNF-α, IL-1β, and other cytokines [55,56,57]. ADAMTS-5 level increases under inflammatory conditions [58]. TIMP-3 is a member of the TIMP family, which are intrinsic moderators of matrix metalloproteinases (MMPs) and crucial for preserving the nearby extracellular matrix [59]. TIMP-3 may have the ability to stop cartilage loss [60,61]. TIMP-3 stands out as a significant regulator of inflammation due to its precision in blocking pro-inflammatory cytokines and damage to joint tissue [62]. Numerous researchers hypothesized that the imbalance between ADAMTS-5 and TIMP-3 may be because of the rapid degradation of the cartilage matrix in RA [63]. In the present study, nano-HAP/Vit B_12_ exhibited a significant reduction of ADAMTS-5 and elevation of TIMP-3 in arthritic rats. Additionally, in this study, nano-HAP/Vit B_12_ was prepared because Vit B_12_ is important for bone health. Previous research has linked Vit B12 deficiency and RA, most likely as a result of deficient nutrition and eventual malabsorption brought on by autoimmune mechanisms [64]. Maintaining healthy amounts of vitamin B_12_, required for cells to protect themselves from the damaging effects of free radicals, might reduce inflammation [65]. According to our findings, nano-HAP/Vit B_12_ had a strong anti-arthritic and anti-inflammatory activity.

TGF-β is a crucial regulator of cellular and physiological processes, such as cell survival, migration, proliferation, differentiation, angiogenesis, and immunosurveillance [66]. It is crucial for the growth and homeostasis of many tissues. In addition to regulating ECM formation and degradation, it also affects cell migration, differentiation, apoptosis, and proliferation. Additionally, these factors regulate immune response and modulate how cells and tissues respond to damage [67]. TGF-β is one of the most well-known immunosuppressive cytokines that is released by practically all immune cells, including T, B, dendritic cells, macrophages, and fibroblasts, and it has a role in immune response [68] and plays a crucial role in preventing abnormal reactions that result in autoimmunity [69]. Increased TGF-β expression in RA is parallel with Lu et al. [70] and Raafat et al. [71]. Since several studies have confirmed the presence of TGF-β in the synovial tissues and synovial fluids of patients with RA, it has been suggested that TGF-β plays a role in the pathogenesis of RA [72]. TGF-β can increase inflammation and joint damage in RA by causing synovial fibroblasts and releasing several pro-inflammatory cytokines such as TNF-α, and IL-1β [73]. Treatment with nano-HAP/Vit B_12_ reduced TGF-β expression. In support of this attribution, it was stated by Soriente et al. that nano-HAP was able to decrease inflammation by reducing the pro-inflammatory cytokine TGF-β [74]. According to our findings, nano-HAP/Vit B_12_ had a strong anti-arthritic and anti-inflammatory activity.

Histopathological examination demonstrated that the synovium of CFA-induced arthritic rats displayed hyperplasia, substantial inflammatory cell infiltration, and substantial cartilage destruction. These results are in harmony with Logashina et al. [75] and Shaaban et al. [76]. Rats treated with nano-HAP/Vit B_12_ showed reduced cartilage deterioration, suppressed pannus development, and decreased synovitis. Therefore, our results clearly showed that nano-HAP/Vit B_12_ is an effective inhibitor of inflammatory responses inside tissues, as well as in regulating inflammatory responses and paw edema induced by arthritis. These results also showed the anti-inflammatory activity of nano-HAP/Vit B_12_ as a therapeutic agent for RA due to its effects in repressing inflammation inside tissues and reducing the swelling features of arthritis. Overall, this study demonstrated the effectiveness of nano-HAP/Vit B_12_ as an anti-arthritic and anti-inflammatory treatment for RA.

## 4. Materials and Methods

### 4.1. Chemicals

CFA was purchased from Sigma Chemical Co., St Louis, MO, USA. Vitamin B12 was obtained from Pharma Swede Pharmaceutical Company, Cairo, Egypt. Other compounds were of analytical grade.

### 4.2. Preparation of Nano-HAP and Nano-HAP/Vit B_12_

HAP nanoparticles were prepared following a previous report by Chen et al. [77]. Briefly, a solution of Ca(NO_3_)_2_ (0.167 mol) and (NH_4_)_2_HPO_4_ (0.1 mol) in distilled water was adjusted to approximately the pH of 3.0 and volume of 1 L using urea (1 mol) as a buffer, diluted nitric acid, and distilled water. Then, the mixture was moved to a hydrothermal vessel made of stainless steel and Teflon, which was heated for 24 h at 95 °C. The precipitate was then filtered, thoroughly washed, dried, and kept for further use.

For the synthesis of nano-HAP/Vit B_12_, Vit B_12_ solution (1000 ppm) was prepared by dissolving in DDW water/(DMSO) dimethylsulfoxide (1:1 volume %). The prepared VB_12_ solution was then mixed with 300 mg of HAP and stored in the dark at room temperature for 24 h. The resultant suspension was centrifuged at 13,000 rpm for 30 min. Vit B_12_-loaded HAP particles were washed with DDW water and dried under vacuum for 12 h at 60 °C. The supernatant from centrifugation was collected and filtered using 0.45 micron nylon filters, and then Vit B_12_ concentration was determined by a UV–VIS spectrophotometer at 360 nm [78] to calculate Vit B_12_ loading efficiency. The amount of VB_12_ loaded onto the HA NPs was determined by a quantitative spectrophotometric method. VB_12_ has maximum absorption at 360 nm. A calibration curve was prepared by measuring the absorbance at 360 nm of standard solutions of VB_12_ (5–50 ppm) using a UV–VIS spectrophotometer.

### 4.3. Characterization of Nano-HAP and Nano-HAP/Vit B_12_

The prepared nano-HAP and nano-HAP/Vit B12 were evaluated using XRD (PANalytical Empyrean, Malvern, United Kingdom) using a scan angle range of 5 to 80 degrees, a scan step of 0.05 degrees, a current of 30 mA, and a 40 KV accelerating voltage. FTIR was acquired using a Bruker vertex 70 FTIR-FT Raman. Scanning electron microscopy and EDX analysis for morphological examination and ascertaining elemental composition were performed using SEM-EDX Quanta FEG250 (Elecmi, Zaragoza, Spain). An automatic surface analyzer employing the N_2_ adsorption–desorption method was used to assess the BET-specific surface area, pore volume, and pore size distribution of nano-HAP and nano-HAP/Vit B_12_ (TriStar II 3020, Micromeritics, Norcross, GA, USA). For the determination of microstructures, HRTEM (high-resolution transmission electron microscopy) (JEOL-JEM 2100, Akishima, Tokyo, Japan) was used.

### 4.4. Molecular Modeling

Monte Carlo simulation was performed using an adsorption locator in BIOVIA Materials Studio 2020. The HAP structure from the materials project was used [79]. The geometry of Vit B_12_ loaded onto the HAP (0 1 0) surface was obtained using an adsorption locator employing the Metropolis Monte Carlo simulation with a simulated annealing method. The Bravais lattice of the HAP structure was triclinic. The Vit B_12_ and HUE were optimized using a Forcite module. The optimized α, β, and γ angles were 90°, 119.990°, and 90°. Lattice lengths of A, B, and C were 9.50281 Ǻ, 6.93592 Ǻ, and 18.9932 Ǻ, respectively. A supercell with 6 × 3 × 3 of this crystal and (0 1 0) surface was cleaved with a 60 Ǻ vacuum above the surface. The COMPASSIII forcefield was used [80]. The electrostatic interaction was calculated using the Ewald method. Atom-based was used to calculate the van der Waals interaction.

### 4.5. Experimental Animals

A total of 30 male Wistar rats weighing 120–150 g were obtained from the Egyptian Organization for Biological Products and Vaccines (VACSERA) (Helwan, Cairo, Egypt). Prior to the experiment, animals were monitored for two weeks to assure that they were infection free. Rats were housed in precisely calibrated polypropylene cages and kept in climate-controlled environments (55.5% humidity levels at RT: 20–25 °C), with an *ad libitum* diet and water and a daily light cycle (10–12 h/day). The experimental work was approved by the Faculty of Science Experimental Animal Ethics Committee (ethical approval number: 022-387), Beni-Suef University, Egypt.

### 4.6. Induction of Arthritis and Animal Grouping

For arthritis induction, 0.1 mL of CFA solution was intrapedally injected into the right hind paw footpad once a day for two days [33]. Following RA induction, the 30 Wistar rats were split into three groups (ten rats per each group) that were treated as shown in Figure 10.

Group 1: (Normal negative control rats): Normal rats in this group were orally given an equivalent amount of vehicle (saline) every day for 21 days.

Group 2: (Arthritic positive control group): CFA-induced arthritic rats were treated for 21 days with equivalent amounts of vehicle (saline) taken orally every day.

Group 3: (Arthritic + Nano-HAP/Vit B_12_): CFA-induced arthritic rats were given a safe oral dose of nano-HAP/Vit B_12_ at a daily dose of 30 mg/kg body weight [81] for 21 days.

### 4.7. Assessment of Paw Edema

In all groups, paw edema, swelling rate, and paw volume were evaluated to monitor the course of arthritis. Rats were anesthetized using 1:2:3 ACE mixtures of alcohol, chloroform, and ether for measurement. Day 0 was the first day of CFA injection, and measurements were taken on Days 3, 7, 14, and 21 post-arthritis induction. The volume of the hind paw was calculated using an electronic caliper.

### 4.8. Collection of Blood and Tissue

After mild anesthesia with diethyl ether, animals were sacrificed on the 21st day. Blood samples were collected and centrifuged (3000 rpm for 15 min), and sera were collected in sterile tubes and stored at −20°C for measurement of RF, CRP, 1L-1β, 1L-17, 1L-4, TNF-α, ADAMTS-5, and TIMP-3 levels. The posterior ankle joints of rats’ right leg from each group were removed. Phosphate buffer was mixed with liver tissue samples before centrifugation (3000 rpm for 15 min at 4 °C). Supernatants were kept at −30 °C until used for investigation of antioxidant activities and oxidative stress. Ankle joint tissues were stored at −70 °C and utilized for molecular investigations. After one day of fixation in 10% neutral buffered formalin, a third tissue piece was prepared for cutting into sections and stained for histological analysis.

### 4.9. Biochemical Investigations

An ELISA kit bought from R&D Systems (Minneapolis, MN, USA) was used in accordance with the manufacturer’s instructions. The levels of serum RF, CRP, 1L-1β, 1L-17, 1L-4, TNF-α, ADAMTS-5, and TIMP-3 were assessed.

### 4.10. Biomarkers of the Antioxidant Defense System and Oxidative Stress Measurement

Malondialdehyde (MDA), glutathione-S-transferase (GST), and reduced glutathione (GSH) were analyzed using kits from Bio-Diagnostic (Dokki, Giza, Egypt).

### 4.11. Histopathological Investigation

On the 21st day of post-arthritis induction, rats were sacrificed, and the posterior ankle joints of the right leg were removed and placed in 10% buffered formalin for 48 h. Bones were decalcified using 10% formic acid for two weeks changing the solution twice a week. A surgical blade was used to determine when the decalcification process was complete. Following decalcification, tissues were washed with PBS, dried with ethanol, and embedded in paraffin wax. Sagittal slices (5 mm thick) were created and stained with hematoxylin and eosin (H&E). A blind histological examination was performed by the center for pathology, including synovitis, cartilage, and bone damage. Sections were classified for cartilage degeneration, bone erosion, synovial hyperplasia (pannus development), and inflammation (infiltration of mononuclear cells) using the system described by Sancho et al. [82]. Each characteristic was rated from 0 to 3, with 0 representing normal, 1 (+) representing mild inflammation, 2 (++) representing moderate inflammation, and 3 (+++) representing severe inflammation. Blood vessels, inflammatory infiltrates, articular cartilage, Pannus, and Menisci were found in the joint cavity space.

### 4.12. q-RT-PCR Analysis

Using Trizol (Invitrogen; Life Technologies, Waltham, MA, USA), total RNA from the tissue was isolated employing Sthoeger et al.’s method [83]. Following the isolation procedure and in line with the manufacturer’s recommendations, the 260:280 ratios were assessed to evaluate RNA’s purity. The production of complementary DNA (cDNA) was carried out using a high-capacity cDNA reverse transcription kit (Applied Biosystems, USA). Used primers are displayed in Table 2. Using the 2^−ΔΔCT^ method, the expression of TGF-β was normalized to the expression of rat GAPDH.

### 4.13. Statistical Analysis

Results are presented as mean ± standard error (SE). The statistical variances across the groups were compared using a one-way analysis of variance (ANOVA). Using SPSS version 20 software, Duncan’s approach for post-hoc analysis compares various groups with significance set at *p* < 0.05.

## 5. Conclusions

The current study revealed that the novel synthesized nano-HAP/Vit B_12_ have promising anti-arthritic, anti-inflammatory, and antioxidant effects on CFA-induced arthritic rats. They caused oxidative stress reduction, inflammation suppression, and enhancement of the antioxidant defense system; additionally, they increased anti-inflammatory markers such as TIMP-3 and IL-4 and decreased inflammatory markers, such as IL-1β, TNF-α, IL-17, ADAMTS-5, and TGF-β levels. All these observed effects indicate the contribution of nano-HAP/Vit B_12_ as an anti-arthritic agent (Figure 11). Furthermore, nano-HAP/Vit B_12_ improved the histopathologic implications of CFA-induced arthritic rats’ articular joints. Finally, we can say that nano-HAP/Vit B_12_ might serve as an effective treatment for RA. More clinical investigations are needed to determine the efficacy and safety of nano-HAP/Vit B_12_.

## Figures and Tables

**Figure 1 pharmaceuticals-16-00551-f001:**
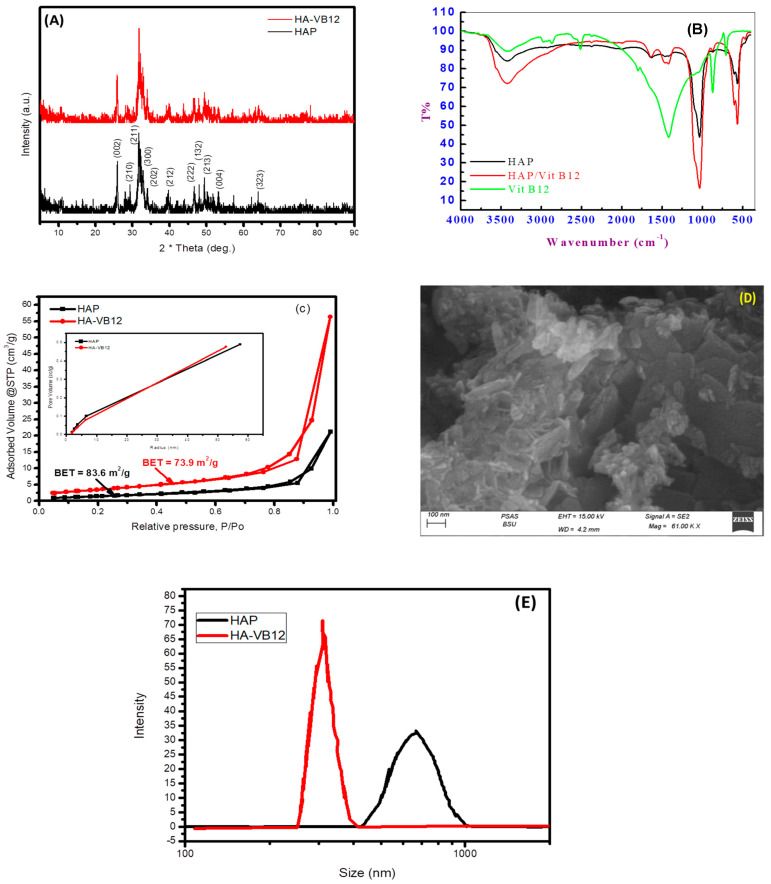
(**A**) XRD; (**B**) FTIR; (**C**) N_2_ adsorption–desorption isotherm (inset: BJH pore size distribution); (**D**) SEM image of the prepared nano-HAP/Vit B_12_ and (**E**) Hydrodynamic size of nano-HAP and nano-Hap/Vit B_12_.

**Figure 2 pharmaceuticals-16-00551-f002:**
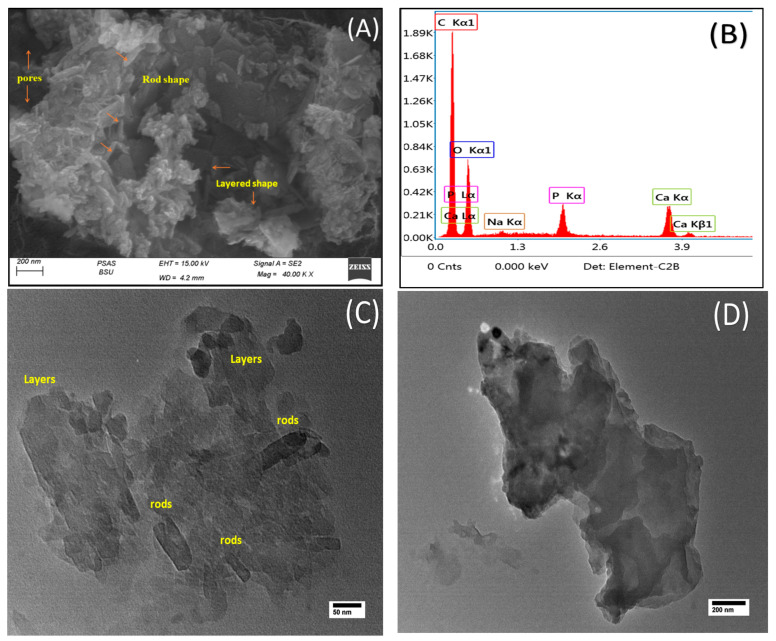
(**A**) SEM; (**B**) EDX; (**C**) and (**D**) TEM images of nano-HAP/Vit B_12_.

**Figure 3 pharmaceuticals-16-00551-f003:**
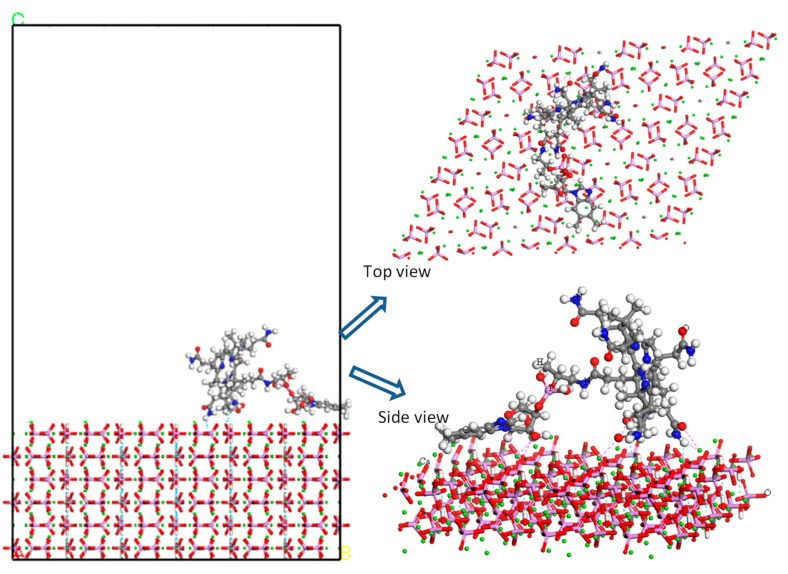
Adsorption of Vit B_12_ on the (0 1 0) HAP surface as obtained from the Monte Carlo simulation.

**Figure 4 pharmaceuticals-16-00551-f004:**
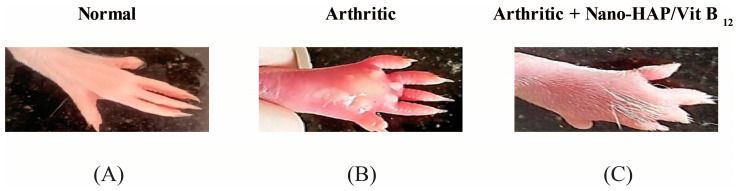
Gross morphology of rats in various groups: (**A**) normal rats; (**B**) arthritic rats; (**C**) arthritic rats receiving nano-HAP/Vit B_12_.

**Figure 5 pharmaceuticals-16-00551-f005:**
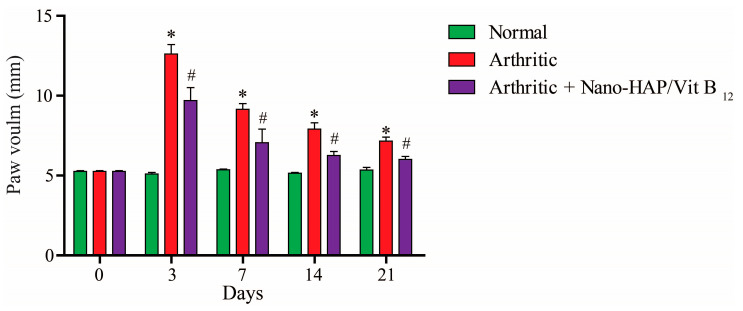
Effect of Nano-HAP/Vit B_12_ on right hind paw size in CFA-induced arthritic rats (symbols indicate significant difference at *p* < 0.05). * Significant compare to arthritic rats, # significant compare to arthritic rats treated with Nano-HAP/Vit B_12_.

**Figure 6 pharmaceuticals-16-00551-f006:**
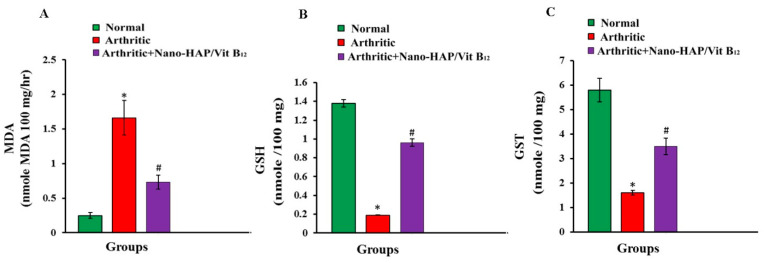
Effect of Nano-HAP/Vit B_12_ on (**A**): LPO, (**B**): GSH content, and (**C**): GST activities. Data are expressed as mean values ± SEM (* *p* < 0.05: significant difference relative to arthritic group, # *p* < 0.05: significant difference relative to arthritic rats treated with Nano-HAP/Vit B_12_).

**Figure 7 pharmaceuticals-16-00551-f007:**
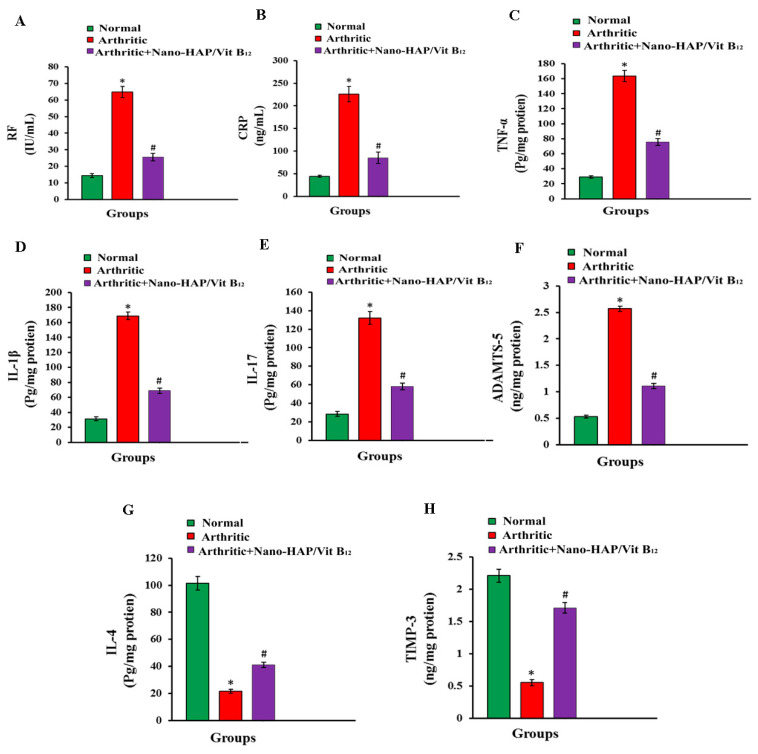
Effects of nano HAP/Vit B_12_ on serum levels of: (**A**) RF; (**B**) CRP; (**C**) TNF-α; (**D**) IL-1β; (**E**) IL-17; (**F**) ADAMTS-5; (**G**) IL-4; (**H**) TIMP-3 (Data are expressed as mean values ± SEM, (* *p* < 0.05: significant difference relative to arthritic rats, # *p* < 0.05: significant difference relative to arthritic rats treated with Nano-HAP/Vit B_12_).

**Figure 8 pharmaceuticals-16-00551-f008:**
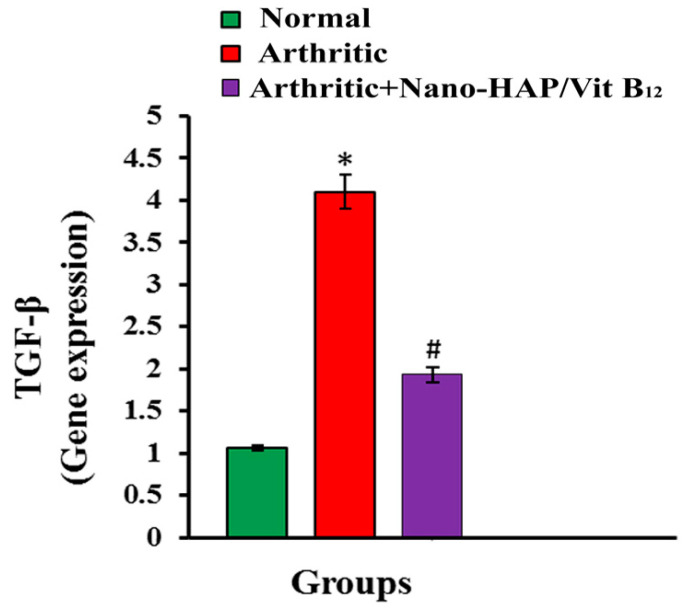
Effect of nano-HAP/Vit B_12_ on TGF-β mRNA expression (Data are expressed as mean values ± SEM, (* *p* < 0.05: significant difference relative to arthritic rats, # *p* < 0.05: significant difference relative to arthritic rats treated with Nano-HAP/Vit B_12_).

**Figure 9 pharmaceuticals-16-00551-f009:**
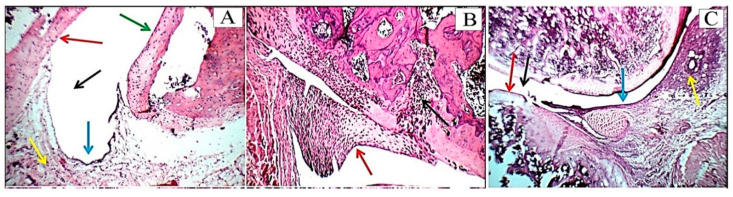
(**A**) In normal rats: high-power view showing average joint cavity (black arrow), average synovial lining (blue arrow), average sub-synovial tissue (yellow arrow), average articular cartilage with intact superficial layer (red arrow), and average menisci (green arrow) (H&E X 200). (**B**) In arthritic rats: high-power view showing destructed Pannus on and on menisci (red arrow), and marked sub-synovial inflammatory infiltrate (yellow arrow) (H&E X 200). (**C**) In nano-HAP/Vit B_12_-treated rats: joint showing narrow joint cavity (black arrow), mildly destructed articular cartilage (red arrow), and intact synovial lining (blue arrow) with marked sub-synovial inflammatory infiltrate (yellow arrow) (H&E X 100).

**Figure 10 pharmaceuticals-16-00551-f010:**
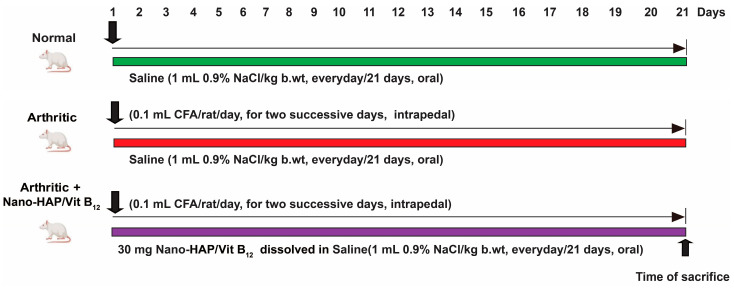
Experimental design and animal grouping.

**Figure 11 pharmaceuticals-16-00551-f011:**
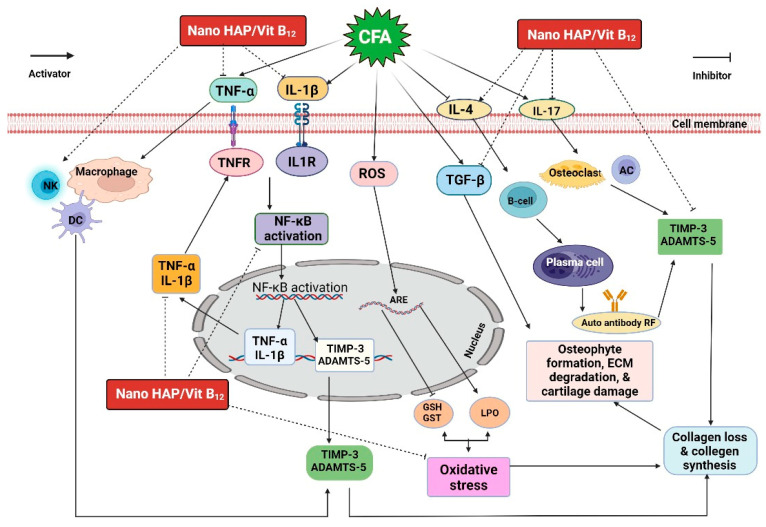
Schematic figure illustrating mechanisms of actions of nano-HAP/Vit B_12_ in arthritic rats.

**Table 1 pharmaceuticals-16-00551-t001:** Ankle histological lesion scores in normal, arthritic, and arthritic treated with nano-HAP/Vit B_12_ rats’ groups.

	Cavity	Synovial Lining	Inflammatory Infiltrate	Blood Vessels	Pannus	Articular Cartilage	Menisci
Normal	0	0	0	0	0	0	0
Arthritic	0	++	+++	0	++	+++	++
Arthritic + nano-HAP/Vit B_12_	+	0	+	0	+	0	0
• Cavity:							
0: Average			+: Narrow			++: Very narrow	
• Synovial lining:							
0: Average/intact			+: Thickened/hyperplastic			+++: Necrotic/ulcerated	
• Inflammatory infiltrate:							
0: No			+: Scattered/mild			+++: Moderate/marked/ with excess fibroblasts	
• Blood vessels:							
0: Average			+: Mildly dilated/congested			++: Markedly dilated/congested	
• Pannus:							
0: No			+: Small/large non-destructing			++: Large destructing/with fibrous bands	
• Articular cartilage:							
0: Average			+: Mildly destructed/thickened			+++: Markedly and severe destructed	
• Menisci:							
0: Average			+: Mildly destructed			++: Markedly destructed	

**Table 2 pharmaceuticals-16-00551-t002:** Primer sequences for rats.

Genes	GenBank Accession Number	Sequence (5′–3′)
TGF-β	XM_032894155.1	F: GACTCTCCACCTGCAAGACC
		R: GGACTGGCGAGCCTTAGTTT
GAPDH	XM_017592435.1	F: CACCCTGTTGCTGTAGCCATATTC
		R: GACATCAAGAAGGTGGTGAAGCAG

## Data Availability

All data can be found in the article.

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
