# Peer review of "A Novel Hydroxyapatite/Vitamin B12 Nanoformula for Treatment of Bone Damage: Preparation, Characterization, and Anti-Arthritic, Anti-Inflammatory, and Antioxidant Activities in Chemically Induced Arthritic Rats"

_pharmaceuticals, 2023, doi:10.3390/ph16040551_

Round 1

Reviewer 1 Report

The work is complete and accepted according to the form presented

Author Response

Reviewer 1:

  1. The work is complete and accepted according to the form presented

Answer: Thank you for your comment and we highly appreciate your feedback.

Reviewer 2 Report

Zaky and co-workers report on an interesting contribution for fumulating "A Novel Hydroxyapatite/Vitamin B12 Nanoformula: Preparation, Characterization, and Anti-arthritic, Anti-inflammatory, and Antioxidant Activities in Chemically Induced Arthritic Rats". 

In general, the reported results and conclusions are interesting, supported by sufficient experimental evidence and seem to be scientifically correct. All in all, I would recommend the publication of this contribution in Pharmaceuticals after a careful consideration on the following. My major concern and critic is related to the usage of English. The report has a lot of grammatical mistakes and typos that must be carefully revised (and many parts re-written) to bring it to an acceptable level for a scientific publication. I also could speculate that the manuscript was written by different persons and there are, at least, two different written styles. While the Introduction part reads, in general, quite well and there are a few typos/grammatical errors, the other sections of the manuscript (Results, Discussion and Conclusion) need to be revised carefully and (some parts) re-written in order to facilitate and optimize the reading of this contribution.

After this major improvement, I would probably be able to provide more feedback or questions in terms of the scientific content of this work.

Author Response

Answer: Answer: Thank you for your comment. All grammar mistakes and typo errors have been corrected and highlighted in red in the revised manuscript. Also, the manuscript was subjected to English editing Editing_ INQ-8256152921_INQ-6656153621_ EKBEAW-3381). Please see the English editing certificate at the end of the letter.

Reviewer 3 Report

The manuscript concerns the use of a simple system based on hydroxyapatite and vitamin B12 in the treatment of rheumatoid arthritis. The results are interesting but some improvements are required:

1. many abbreviations are not explained in the manuscript (including the abstract and the main text);

2. please carefully review the entire text for incorrect spelling of degrees, chemical formulas, etc. (upper and lower index);

3. FTIR analysis: I'm not convinced with the interpretation for HA-VB12: where are the signals from vitamin? additionally, Vit B12 spectrum should also be included for comparison and discussion;

4. the quality of Figs 1& 2 is poor;

5. XRD pattern: what is the number of HAP patterns - it should be included;

6. please, specify the amount of adsorbed Vit B12; what was the optimized amount of the vitamin? 

7. what is the novelty of the paper; please compare your results with other studies concerning HAB application in rheumatoid arthritis.;

8. Abstract, line 26: I suppose, it should be "particle size", not "practical size".

Author Response

Reviewer 3:

  1. Many abbreviations are not explained in the manuscript (including the abstract and the main text).

Thanks for your comment. All the abbreviations have been explained in the manuscript and highlighted in red in the revised version.

  1. Please carefully review the entire text for incorrect spelling of degrees, chemical formulas, etc. (upper and lower index).

Thanks for your comment. All the incorrect spelling of degrees, chemicals, and formulas have been revised in the manuscript and highlighted in red in the revised version.

  1. FTIR analysis: I'm not convinced with the interpretation for HA-VB12: where are the signals from vitamin? Additionally, Vit B12 spectrum should also be included for comparison and discussion.

Thanks for your comment. A new FTIR figure contains the signals of the vitamin was added in the revised manuscript. Also, the interpretation of the FTIR analysis was revised and highlighted with red in the revised manuscript.

  1. The quality of Figs 1& 2 is poor.

Thanks for your comment. The quality of Figures 1 and 2 were improved in the revised version.

  1. XRD pattern: What is the number of HAP patterns - it should be included?

Thanks for your comment. The numbers of HAP patterns were added in the revised version and highlighted with red in the revised manuscript. The numbers were added as fellows:

Please see page 3 lines 114 to 120

        Signed Spectra of Vit B12 (Fig. 1B) at 3420.05, 2974.93, 2374.66, 1801.08, 1420.70, 873.93 and 708.90 cm-1. Band assigned at 2974.93 cm−1 is characteristic of stretching of C-H, band assigned at 1801.01 cm−1 is characteristic of stretching of C=C, 1420.70 cm-1 for C=O and 873.93 and 709.90 cm-1 for C-O. appearing of some peaks related to Vit B12 in the HAP/Vit B12 like 1432.87 and 875.13 cm-1 conforming the loading of the Vit B12, Also  broadening in peak at 3421.63 cm-1 that conform increasing in H-Bond. 

  1. Please, specify the amount of adsorbed Vit B12; what was the optimized amount of the vitamin? 

      Thanks for your comment.  The amount of VB12 loaded into the HAP NPs was determined by a quantitative spectrophotometric method. VB12 has maximum absorption at 360 nm. A calibration curve was prepared by measuring the absorbance at 360 nm of standard solutions of VB12 (5–50 ppm) using a UV-Vis spectrophotometer. VB12-loaded HA particles were washed with DDW water and dried under vacuum for 12 h at 60 oC. The Vit B12 solution, supernatant was collected. Vit B12 concentration was detected and measured by UV-viz spectrophotometric at 360 nm to determine the Vit B12 loading efficiency in nano HAP. The solution was filtered after centrifugation by using 0.45 mm nylon filters before spectrophotometric analysis, please see the the calculation:

  • Loaded drug (mg/g) = (Initial concentration − Residual concentration) × solvent volume/ Carrier weight (2)=(690-200)*100*.001/0.1=49mg/g, Loading efficiency=(690-200)*100/690=71.01%
  • Calibration curve:

Y = A + B * X

Parameter             Value      Error

------------------------------------------------------------

A                -1E-4                    6.35085E-4

B                  4.7E-4 1.91485E-5

R                             SD                                        N       P

  0.99752               6.0553E-4              5              1.4825E-4

  1. What is the novelty of the paper; please compare your results with other studies concerning HAB application in rheumatoid arthritis

       Thanks for your comment. We compared our results with those of other studies in the revised version. The novelty of our study is addressed below.

       Previous research found that Nano HAP got into bone tissues and gave them the calcium phosphates they needed to heal after the disease had hurt them [3, 19, 20]. By high-lighting this gap in the introduction, our study revealed its uniqueness by combining HAP and Vit. B12, which resulted in a useful nonformula and was helpful in treating rheumatoid arthritis.

  1. Ain, Q.; Zeeshan, M.; Khan, S.; Ali, H. Biomimetic hydroxyapatite as potential polymeric nanocarrier for the treatment of rheumatoid arthritis. J. Biomed. Materials Res. 2019, 107(12), 2595-2600.
  2. Weissig, V.; Pettinger, T.K.; Murdock, N. Nanopharmaceuticals (part 1): products on the market. Intern. J. nanomed. 2014, 9, 4357–4373.
  3. Nabipour, Z.; Nourbakhsh, M.S.; Baniasadi, M. Evaluation of ibuprofen release from gelatin/hydroxyapatite/polylactic acid nanocomposites. Iranian J. Pharmaceut. Sci. 2018, 14, 75–84.

Our study revealed new observations disproving an existing idea and qualifies as a novelty as when arthritic rats were given Nano HAP/Vitamin B12, the increased hind paw size compared to arthritic rats was reduced, which could be attributed to Nano HAP/Vitamin B12's anti-inflammatory effects. We have measured malondialdehyde (MDA) level, glutathione-S-transferase (GST) activity, and GSH content and found that the Nano HAP and Vit B12 had a strong antioxidant activity against the ROS produced by CFA induction.

- It was stated by Soriente et al. (2021) that Nano HAP lonely was able to decrease inflammation by reduced pro-inflammatory cytokine TGF-β, while in our study we treated by Nano HAP/Vit B12 that have shown the novelty of your study by Using the 2−ΔΔCT method indicating that treatment with Nano HAP/Vit B12 was heplful to reduce the expression of TGF-β and add to the existing literature that Nano HAP/Vit B12 had a strong anti-arthritic and anti-inflammatory.

  1. Soriente, A.; Amodio, S.P.; Fasolino, I.; Raucci, M.G.; Demitri, C.; Engel, E.; Ambrosio, L. Chitosan/PEGDA based scaffolds as bioinspired materials to control in vitro angiogenesis. Materials Sci. Eng. 2021, 118, 111420.

- In conclusion, our study advances the knowledge in the field of Nano HAP/Vit B12 application in rheumatoid arthritis as  our results also showed the anti-inflammatory activity of Nano HAP/Vit B12 as therapeutic targets for RA due to their inhibitory effects in repressing inflammation inside tissues and reducing the swelling features of arthritis.

  1. Abstract, line 26: I suppose, it should be "particle size", not "practical size".

Thanks for your comment. Particle size was revised in the revised version and highlighted with red in the revised manuscript.

Round 2

Reviewer 2 Report

I am satisfied with the implemented changes by the authors of this contribution. As mentioned before, I believe that this contribution provides enough experimental evidence to scientifically support its claims, whereas the utilized experimental and analysis methods seem to be appropriate. Thus, I would recommend the publication of this work in Pharmaceuticals.

However, I would recommend the authors to carefully revise one more time the entire manuscript as I could still detect some typos,  punctuation errors, etc. in this revised version:

Line 95: Should read "illustrated"

Line 118: ". appearing" please correct; "like" change to "such as"?

Line 119: ", Also   broadening" please correct.

Line 133; "particles" should read "particle"

Line 267: "existence of carbonate"?

Line 269: Add a "," after "ions"

Line 281: "Theis" please correct.

Line 309: "suggeting" please correct.

Line 405: "mm" should be "micron"?

Line 407: ".   ." please correct.

I would also suggest the authors to define al the utilized acronyms as they appear for the first time in the manuscript, including the abstract section.

Author Response

Line 95: Should read "illustrated"

Answer: Thanks for this comment. It has been corrected and highlighted in red in the revised manuscript.

Line 118: ". appearing" please correct; "like" change to "such as"?

Answer: Thanks for this comment. It has been corrected and highlighted in red in the revised manuscript.

Line 119: ", Also   broadening" please correct.

Answer: Thanks for this comment. It has been corrected and highlighted in red in the revised manuscript.

Line 133; "particles" should read "particle"

Answer: Thanks for this comment. It has been corrected and highlighted in red in the revised manuscript.

Line 267: "existence of carbonate"?

Answer: Thanks for this comment. It has been corrected and highlighted in red in the revised manuscript.

Line 269: Add a "," after "ions"

Answer: Thanks for this comment. It has been corrected and highlighted in red in the revised manuscript.

Line 281: "Theis" please correct.

Answer: Thanks for this comment. It has been corrected and highlighted in red in the revised manuscript.

Line 309: "suggeting" please correct.

Answer: Thanks for this comment. It has been corrected and highlighted in red in the revised manuscript.

Line 405: "mm" should be "micron"?

Answer: Thanks for this comment. It has been corrected and highlighted in red in the revised manuscript.

Line 407: ".   ." please correct.

Answer: Thanks for this comment. It has been corrected and highlighted in red in the revised manuscript.

I would also suggest the authors to define all the utilized acronyms as they appear for the first time in the manuscript, including the abstract section.

Answer: Thanks for this comment. It has been corrected and highlighted in red in the revised manuscript.

Reviewer 3 Report

The Authors followed all my comments.

Author Response

The Authors followed all my comments.

Answer: Thanks for your valuable comment and feedback.